# Machine Learning Techniques for Matching Candidates' Profiles with Job Advertisements

Aleksandra Davidova Stefanova and Tosin Adewumi

Luleå University of Technology, Department of Computer Science, Electrical and Space Engineering
{setale-4@student.ltu.se, tosin.adewumi@ltu.se}

## Abstract

This study presents a comparison between machine learning models and traditional keyword matching algorithms in resume-job matching tasks using a public dataset from Kaggle and a private corporate dataset. The results showed that the Ada and T5-based models generated the highest cosine similarity scores, which align with human judgment. This suggests that traditional keyword matching algorithms are not as well-suited for the task as much as state-of-the-art (SotA) deep learning models.

## 1 Introduction

Online recruitment plays a crucial role in connecting job seekers with potential employers. The growing amount of online recruitment data makes conventional manual selection cumbersome and highly ineffective. According to a report from LinkedIn, there were 20 million job listings on LinkedIn and 660 million users from over 200 countries and territories all over the world as of late November 2019 [1]. It has been estimated that between 60% to 75% of the resumes never get through the applicant tracking system (ATS) before even reaching a recruiter [2]. Traditional keyword matching algorithms were only partially effective as they often failed to capture the overall context of resumes and struggled to interpret CVs when the exact job advertisement keywords were sent from the resumes [3]. Machine learning techniques are gaining increasing popularity among talent acquisition companies and divisions [2] as they are able to evaluate resumes based on their content and context, aligning them more precisely with the job description. This research have focused on answering the following research question:

**RQ:** Which machine learning techniques (e.g., similarity-based methods, deep learning embeddings, natural language processing) provide the most effective matching between candidate profiles and job advertisements, and under what circumstances do certain techniques outperform others?

## 2 Method

All models were tested using a public dataset from Kaggle [4] containing pre-matched job-resume pairs and a private corporate dataset provided by a Swedish digital recruitment and talent attraction company. Both datasets were preprocessed to prepare them for resume-job matching. All resumes and jobs were encoded into dense vector embeddings using ten types of transformer models and one traditional keyword-based matching method (TF-IDF). The transformer models were developed and maintained by the Ubiquitous Knowledge Processing (UKP) Lab [5] and are available on HuggingFace [6]. Cosine similarity was applied afterwards to evaluate how similar each resume is to the different job postings and a few cases humanly validated. Each evaluation was run five times to obtain the mean cosine similarity score (and standard deviation).

## 3 Results and Discussion

Results, as indicated in Tables 1 and 2, show that the sequence-to-sequence transformer model, "sentence-t5-base" [7], and the OpenAI encoder-only embedding model - "text-embedding-ada-002" (Ada-002) [8], generated the highest cosine similarity scores, aligning more with human judgment. The former achieved a score of 0.765 and the latter a score of 0.771, respectively, on the Kaggle dataset while having 0.768 and 0.759, respectively, on the private dataset. The low standard deviations (SDs) (0.001 and 0.003) show that the models' results do not fluctuate much across different splits and samples, which suggests that they are more stable and reliable. All of the Transformer models captured semantic similarities in the job resume pairs as they mapped sentences into a dense semantic space where cosine similarity reflected semantic closeness. Unlike TF-IDF, they did not match only exact word overlap in resume-job pairs. This resulted in more bell-shaped distributions, centered around higher scores, because semantically related sentences cluster closer together.

The next set of models, all-MiniLM-L6-v2 [9], all-MiniLM-L12-v2 [10], multi-qa-MiniLM-L6-cos-v1 [11], multi-qa-distilbert-cos-v1 [12], and all-

**Table 1.** Average cosine similarity results and standard deviation for the Kaggle dataset.

| Model | Mean | SD |
|---|---|---|
| **sentence-t5-base** | **0.765** | **0.003** |
| **text-embedding-ada-002** | **0.771** | **0.003** |
| all-MiniLM-L6-v2 | 0.204 | 0.013 |
| all-MiniLM-L12-v2 | 0.247 | 0.013 |
| multi-qa-MiniLM-L6-cos-v1 | 0.171 | 0.017 |
| multi-qa-distilbert-cos-v1 | 0.228 | 0.007 |
| all-mpnet-base-v2 | 0.256 | 0.013 |
| all-distilroberta-v1 | 0.140 | 0.012 |
| paraphrase-mpnet-base-v2 | 0.189 | 0.014 |
| gtr-t5-base | 0.523 | 0.011 |
| TF-IDF | 0.035 | 0.003 |

**Table 2.** Average cosine similarity results and standard deviation for the private dataset.

| Model | Mean | SD |
|---|---|---|
| **sentence-t5-base** | **0.768** | **0.001** |
| **text-embedding-ada-002** | **0.759** | **0.001** |
| all-MiniLM-L6-v2 | 0.206 | 0.005 |
| all-MiniLM-L12-v2 | 0.248 | 0.007 |
| multi-qa-MiniLM-L6-cos-v1 | 0.186 | 0.003 |
| multi-qa-distilbert-cos-v1 | 0.192 | 0.004 |
| all-mpnet-base-v2 | 0.251 | 0.005 |
| all-distilroberta-v1 | 0.195 | 0.004 |
| paraphrase-mpnet-base-v2 | 0.213 | 0.005 |
| gtr-t5-base | 0.535 | 0.003 |
| TF-IDF | 0.008 | 0.001 |

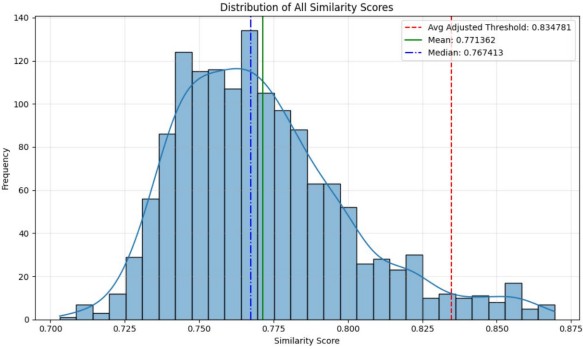

**Figure 1.** Distribution of similarity scores with the semantic embedding model text-embedding-ada-002 tested with the Kaggle dataset.

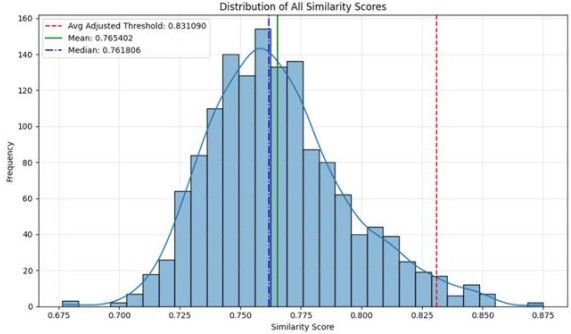

**Figure 2.** Distribution of similarity scores with the sentence_t5_base model tested with the Kaggle dataset.

distilroberta-v1 [13], showed moderate performance as they are smaller, encoder-only, distilled models that are optimized to be efficient and faster at the expense of sacrificing semantic richness. The all-mpnet-base-v2 [14] and the paraphrase-mpnet-base-v2 [15] are MPNet-based, non-distilled encoder-only models that are trained on paraphrasing similarity tasks. They performed moderately well but not as strongly as the T5-based models. The all-mpnet-base-v2 consistently outperformed paraphrase-mpnet-base-v2 as it is more adapted to general embedding training rather than paraphrase detection.

The worst performing model of all was the TF-IDF, which produced near-zero scores for the embedding-based semantic tasks, with a mean of 0.035 for the Kaggle dataset and 0.008 for the private dataset. The model can not understand semantics or contextual meaning and captures only exact keyword overlap. The small SD values indicate that the TF-IDF gave relatively consistent similarity scores rather than sometimes performing well and sometimes poorly across runs. The low mean and SD showed both ineffectiveness in accurate job-resume comparisons and consistent failure across runs, rather than randomness.

## 4 Conclusion

The current research presented a direct comparison between modern ML semantic embedding models and traditional keyword matching algorithms, specifically TF-IDF. All models were tested using a public dataset from Kaggle and a private corporate dataset. The outcomes showed that the T5-based and Ada models captured the semantic similarities best. The research methodology has some limitations that could be a subject of future research. The transformer models could be fine-tuned on the given datasets, so they can fit the data more closely. Other types of ML models, such as large language models (LLMs), and even other traditional keyword matching algorithms could also be compared in future studies, using different datasets.

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
