# OpenReview forum: "Machine Learning Techniques for Matching Candidates’ Profiles  with Job Advertisements"
_NLDL.org/2026/Abstracts_Track — NLDL 2026 Abstracts_

### Official Review · Reviewer_YZZK · 2025-10-24

**Soundness:** 1
**Correctness:** 1
**Rating:** 1
**Confidence:** 4

**Summary:**

The authors present how several sequence encoders align the embeddings of job-postings with resumes. The analysis is performed on a public dataset and a closed in-house dataset. Results of feature alignment is reported in cosine-similarity between job-posting and resume embeddings. They show that the sentence-t5-base and text-embedding-ada-002 models yield higher cosine similarities across both datasets.

**Strengths:**

The authors test the performance across a sufficient number of encoder models. As no training is performed, the size of the datasets (at least the public) should suffice. The authors clearly state the motivation for the research.

**Weaknesses:**

Their is unfortunately a significant lack of novelty here. Several ressources can be found that performs the same analysis on the same data. The conclusion, first mentioned in the abstract that classical algorithms via TF-IDF are not sufficient has been shown several years ago.

The setup is poorly explained, there is no clear indication which embeddings the authors are comparing. Ideally they would compare embeddings between job posts and resumes used to apply for it and compare that with alignment "out-of-group". However, it is not explicitly stated from what i can see. Therefore i fear that they simply compared the average cosine distance between all postings and resumes - which favors models that always predict towards the same direction in embeddingsspace.

Simply comparing cosine similarities is often not sufficient, TF-IDF for example would tend to produce more dissimilar embeddings than the other methods, therefore it is necessary to include some performance measure for a downstream matching algorithm.

There is no explanation of the datasets neither the public nor the in-house one.

---

### Official Review · Reviewer_F6tU · 2025-10-24

**Soundness:** 2
**Correctness:** 2
**Rating:** 2
**Confidence:** 3

**Summary:**

This abstract presents a comparison study between traditional keyword matching (TF-IDF) and modern transformer-based embedding models for resume-job matching. The authors tested 11 different approaches on both a public Kaggle dataset and a private corporate dataset, finding that Ada-002 and sentence-T5-base models significantly outperformed traditional methods.

**Strengths:**

- Resume-job matching is a practical and important application of ML with clear industry relevance.
- The study compares 10 transformer models plus TF-IDF, providing useful benchmarking information for the community.
- Using both public and private datasets strengthens the findings and shows consistency across different data sources.
- Encoding documents into embeddings and using cosine similarity for matching, with multiple runs for stability assessment.
- Tables and figures effectively communicate the performance differences between models.

**Weaknesses:**

Major:
- The discussion mostly describes what happened rather than providing deeper insights into why certain models performed better.

Minor:
- The abstract lacks details about preprocessing steps, dataset sizes, or how "human judgment" validation was conducted.
- Given this is just a comparison, significance testing might have been more meaningful to add.
- The claim about alignment with "human judgment" needs more explanation - how many cases were validated and by whom?

---

### Official Review · Reviewer_NecR · 2025-11-02

**Soundness:** 1
**Correctness:** 2
**Rating:** 2
**Confidence:** 3

**Summary:**

This study provides a comparison of ten transformer-based embedding models and the inclusion of the TF-IDF method. Two job-resume datasets are used to evaluate the aforementioned models, transforming each constituent of the pair into embeddings. These were then scored with cosine similarity. The mean across the respective dataset is used as the final test results. The author's conclusion is that keyword matching algorithms perform worse than contemporary transformer-based embedding models.

**Strengths:**

Overall, the paper is well formatted and clear. The research question and the proposed hypothesis being tested are outlined clearly. Differences between the transformer embedding models and TF-IDF are mentioned.

The usage of two domain-specific dataset sources, combined with 5-meaned runs, is a positive inclusion for the correctness of the proposed paper.

Overall, the central theme is of relevance to the ML community.

**Weaknesses:**

Correctness:
1. The hugging face citation present in Section 2 (citation 6) does not link to the Hugging Face repository mentioned.

Soundness:
1. Human judgement is mentioned but not clarified, making it hard to understand how this impacted results or whether this is simply a qualitative element omitted for the abstract (Section 3). Furthermore, it's unclear what humanly validated implies (Section 2).
2. Within Section 2, all vectors are labelled as dense. However, TF-IDF typically produces sparse embeddings unless dimensionality reduction is applied. As these are not outlined or mentioned, the core methodology is confusing.
3. The core proposition of using cosine similarity to compare model-wise, ten transformers and the TF-IDF method, poses severe issues. While cosine similarity evaluation is present in the literature, to my understanding, there must be steps taken to ensure similar embedding dimensions across models. As such considerations are not mentioned, it's hard to evaluate the soundness of this approach.  The use of TF-IDF, which, unless stated otherwise, is a sparse vector, would likely lead to greatly diminished similarity scores. As this is the core quantitative method, it severely undermines the core contribution of the study.

Other:
1. Section 4, the conclusion mentions some methodology limitations, but this is not elaborated upon.

---

### Decision · Program_Chairs · 2025-11-05

**Decision:**

Accept

**Comment:**

Reviewers focused largely on novelty, which is not required for abstracts. The PCs find the submission relevant to the conference topics and suitable for an engaging poster-session discussion. Because abstracts are non-archival, we encourage discussion to help the authors refine and improve their work.